# Recent Advances in Iron Chelation and Gallium-Based Therapies for Antibiotic Resistant Bacterial Infections

**DOI:** 10.3390/ijms22062876

**Published:** 2021-03-12

**Authors:** Víctor Vinuesa, Michael J. McConnell

**Affiliations:** Intrahospital Infections Laboratory, National Centre for Microbiology, Instituto de Salud Carlos III, 28220 Madrid, Spain; vvinuesa@isciii.es

**Keywords:** iron chelators, multidrug resistant bacteria, gallium, synergy, iron acquisition, DIBI

## Abstract

Iron is essential for multiple bacterial processes and is thus required for host colonization and infection. The antimicrobial activity of multiple iron chelators and gallium-based therapies against different bacterial species has been characterized in preclinical studies. In this review, we provide a synthesis of studies characterizing the antimicrobial activity of the major classes of iron chelators (hydroxamates, aminocarboxylates and hydroxypyridinones) and gallium compounds. Special emphasis is placed on recent in-vitro and in-vivo studies with the novel iron chelator DIBI. Limitations associated with iron chelation and gallium-based therapies are presented, with emphasis on limitations of preclinical models, lack of understanding regarding mechanisms of action, and potential host toxicity. Collectively, these studies demonstrate potential for iron chelators and gallium to be used as antimicrobial agents, particularly in combination with existing antibiotics. Additional studies are needed in order to characterize the activity of these compounds under physiologic conditions and address potential limitations associated with their clinical use as antimicrobial agents.

## 1. Introduction

Antimicrobial resistance is a significant burden on public health. The emergence of antibiotic resistance in multiple bacterial species affects many aspects of modern medicine, from the treatment of infections in the primary care setting to the clinical management of critically-ill patients receiving intensive care. The global dissemination of bacterial strains with resistance to multiple antibiotic classes represents a particular challenge to appropriate therapy, as in some cases there are few clinically-available antimicrobials that retain sufficient activity against these isolates [1]. Reports describing infections caused by pandrug resistant strains with resistance to all clinically-used antibiotics are especially worrisome since treatment options in these cases are severely limited [2]. The global health burden of antibiotic resistance is difficult to estimate. A report commissioned by the government of the United Kingdom in 2014 estimated that global deaths due to antimicrobial resistance could increase to 10 million per year, compared to an estimated 700,000 deaths in 2014, if current trends continue [3]. Although this report included multiple types of antimicrobial resistance and these long term projections are inherently difficult to quantify, the estimated impact of microbial resistance moving forward is clearly a cause for concern. This situation calls for the development of new antimicrobials with novel mechanisms of action. Unfortunately, very few antibiotics with completely novel mechanisms of action have been approved for clinical use over the last 40 years, and so there are currently very few new compounds in the development pipeline [4].

Given the continued emergence and dissemination of antimicrobial resistance, and the low likelihood that the need for new compounds with novel mechanisms of action will be met by the current development pipeline, exploring the potential of alternatives to traditional small molecule antibiotics is warranted. A recent report aiming to comprehensively evaluate alternatives to small molecule antimicrobials at the portfolio level identified metal chelation therapy (zinc, manganese and iron) as one of 19 approaches with potential for the treatment and prevention of antibiotic resistant infections [5]. Over the last decade there has been special interest in targeting bacterial iron metabolism as an antimicrobial strategy given the necessity of iron for the growth and survival of most pathogenic bacterial species. Iron chelation and gallium-based therapies have been broadly studied with multiple different bacterial species, most notably with pathogenic species that are associated with multidrug resistant infections. Studies ranging from the in-vitro characterization of the antimicrobial activity of different chelators to in-vivo studies in animal models of infection have shed light on the potential of iron chelators as potential antimicrobial agents. In this review, we provide a synthesis of published studies that have evaluated the antimicrobial activity of iron chelating molecules and gallium-based therapies on bacterial species associated with antibiotic resistance, with particular emphasis on recent studies with novel compounds, and studies that have evaluated potential synergies with existing antimicrobials. In addition, we provide an overview of the challenges that remain to be addressed before iron chelation therapy could be used clinically. Approaches other than iron chelation and gallium-based therapies that target bacterial iron metabolism, such as inhibiting siderophore biosynthesis and assimilation, antibiotic-siderophore conjugates, and inhibiting heme assimilation have been explored and are reviewed elsewhere [6,7].

## 2. Iron Acquisition in Pathogenic Bacteria

Iron is required by most bacterial pathogens for growth and survival, and is therefore essential during the establishment of host colonization and infection. Many species have evolved to use iron in multiple physiological processes including, DNA synthesis, transcription and cellular respiration, likely due to its high abundance in many natural environments (it is one of the most abundant elements on earth) and its readily exploitable oxidation/reduction chemistry. However, in contrast to natural environments and bacterial growth media, free iron in the human body is maintained at a very low concentration, approximately 10^−24^ M [8], due to potential damage to the host caused by the high reactivity of non-complexed iron. Maintaining free iron at very low concentrations also serves to strengthen host defense against infection by contributing to what has been termed nutritional immunity [9]. Low free iron concentrations in the body are achieved by the activity host proteins that form strong complexes with iron. Approximately 70% of total body iron is found in red blood cells in the form of hemoglobin-bound heme complexes [9]. Ferric (Fe(III)) iron is also sequestered by the host proteins transferrin and lactoferrin, which exhibit very high iron affinities with binding constants of ~10^−20^ M and ~10^−23^ M, respectively [10,11]. Lactoferrin is secreted during the innate immune response and maintains its high affinity or iron in the acidic environment created by infection, thus promoting nutritional immunity [12].

Pathogenic bacteria have evolved mechanisms to obtain host iron that can be broadly classified into two groups; (i) siderophore-mediated iron acquisition, and (ii) specific acquisition mechanisms that obtain iron from complexed host proteins such as heme, transferrin and lactoferrin. These mechanism are briefly summarized here; however, the reader is referred to a number of excellent monographic reviews covering these processes [9,13,14,15]. As a general mechanism, bacterial siderophores are extracellular molecules with very high affinity for ferric iron (with binding constants of 10^50^ M^−1^ in some cases) that are secreted by many different pathogenic Gram positive and Gram negative species (Figure 1). The high affinity of siderophores allows them to strip iron from host proteins, after which the siderophore-iron complex is internalized via a siderophore-specific receptor on the bacterial cell surface [6]. Upon reaching the bacterial cytosol, ferric iron is released to the intracellular iron pool either through cleavage of the siderophore or through the reduction of Fe(III) to Fe(II), which lowers binding affinity to the siderophore thus permitting its dissociation [9]. It is important to note that this general mechanism does not apply to all siderophore systems, as it has been shown that the *Pseudomonas aeruginosa* siderophore pyoverdine releases iron in the periplasm before transport to the cytosol [16].

Acquiring iron from host transferrin and lactoferrin occurs via the interaction between these proteins, complexed with ferric iron, and specific receptors on the surface of bacterial cells. These receptors are able to strip Fe(III) from transferrin and lactoferrin and transport it to the interior of the cell. In the case of heme iron, multiple secreted bacterial effectors have been shown to lyse erythrocytes in order to liberate heme and its associated ferrous (Fe(II)) iron. In addition, many bacterial pathogens secrete hemophores, which capture heme from host hemoglobin and shuttles it to receptors on the bacterial outer membrane [17]. These specific receptors on the bacterial cell surface bind these heme complexes which are then transported to the cytosol. Bacterial enzymes, such as heme monooxygenase, cleave the heme moiety and release the Fe(II) into the intracellular iron pool. The acquisition of ferrous iron can also be facilitated by specific uptake systems, such as the Feo system, which is broadly distributed in different Gram negative species [18].

## 3. Iron Chelators as Antimicrobial Agents

Given the absolute necessity of iron for the growth and survival of many pathogenic microorganisms, decreasing available iron at the site of infection has potential to contribute to treatment approaches. One strategy for achieving iron limitation is the use of chelating molecules that sequester the metal and prevent its uptake by the microorganism causing infection. Iron chelators can bind iron in both its ferrous and ferric states; however, they typically have greater affinity for one or the other [19,20]. In addition, some chelators may also be able to bind other metal ions such as copper(II), zinc(II) or gallium(III). The ability of these compounds to chelate Fe(III) with greater stability and affinity depends on the ligands available for iron coordination. Within chelator-iron complexes, Fe(III) is coordinated by forming an octahedral structure next to six donor atoms in which the metal remains in the center. These ligands can be classified according to whether they have two (bidentate), three (tridentate) or six (hexadentate) donor atoms available for coordination with iron [20]. The stability of the chelator-iron complex improves with more donor atoms available in the ligand, with the hexadentates demonstrating the greatest stability. This may explain why most siderophores produced by microorganisms use hexadentate coordination.

The antimicrobial activity of iron chelators has been studied and demonstrated for decades [21,22]. In the following sections, we summarize published studies evaluating the activity of different iron chelators against the most problematic antibiotic resistant bacterial species. Chelators have been grouped according the chemical moiety that participates in coordination of Fe(III) and include; hydroxamates, aminocarboxylates, hydroxypyridinones and cathecols (Figure 2).

### 3.1. Hydroxamates

Compounds derived from hydroxamic acid (Figure 2A) have high affinity for Fe(III). Many siderophores produced by microorganisms, including the prototypical enterobactin from *Escherichia coli*, contain hydroxamate moieties [8]. One of the first iron chelators to be evaluated as an antimicrobial was deferoxamine (DFO), a hexadentate trihydroxamate siderophore produced by *Streptomyces pilosus* that is FDA-approved for the treatment of thalassemia and other iron disorders. DFO can inhibit the growth of *Staphylococcus aureus*, *Pseudomonas aeruginosa* and *Acinetobacter baumannii* at concentrations between 2.5 and 10 mg/mL in Mueller-Hinton broth, a medium with high iron content (Table 1) [23]. In another study, DFO was tested against these same pathogens, in addition to *Escherichia coli* and *Kleblsiella pneumonia*, with no inhibition observed at concentrations ≤512 mg/L in Mueller-Hinton broth and in iron-poor RPMI 1640 [24]. DFO has also demonstrated efficacy in reducing the formation of biofilm in a cellular model of cystic fibrosis by 42% in *P. aeruginosa* at 400 mg/L (approximate 0.71 mM) [25]. However, in the same work, DFO failed to inhibit established biofilms in these cells and in an abiotic plastic surface at the same concentration [25]. This is in line with a study by Banin et al. [26], which describes no effect of DFO either against biofilm formation (at 0.001 mM) and established biofilms (at 1 mM).

Another natural compound, the siderophore produced by *Mycobacterium smegmatis* exochelin-MS (Exo-MS), has been tested for antimicrobial activity against *S. aureus*, *P. aeruginosa* and *A. baumannii* and was more inhibitory at lower concentrations than DFO (minimum inhibitory concentrations (MICs) between 0.05 and 0.125 mg/mL) [23].

A potential limitation of microorganism-derived compounds is the possibility that other microbial species can recognize these xenosiderophores, internalize them and exploit the associated iron. Different pathogens, such as *P. aeruginosa*, *S. aureus* or *E. coli*, have been shown to be able to use iron-DFO complexes as an iron source [37,38,39,40]. In fact, Visca et al. described the capability of siderophore null mutant *P. aeruginosa* to grow in an iron-poor media at suboptimal concentration of DFO (20 µM) better than without it [41]. Also, in a study by de Leseleuc et al. [42], the growth of *A. baumannii* in RPMI supplemented with 10% fetal bovine serum was encouraged by the presence of DFO. This may explain the high MICs seen with DFO against these microorganisms. However, the ability of DFO to be taken up by bacteria that do not produce it has been revealed as an effective strategy to transport different molecules with antimicrobial activity into these pathogens [26,43].

### 3.2. Aminocarboxylates

Aminocarboxylates (Figure 2B) include a wide variety of compounds known for their chelating capacity, such as ethylenediaminetetraacetic acid (EDTA) and diethylenetriaminepentaacetic acid (DTPA). They have high affinity for Fe(III) but generally low selectivity, which can cause toxicity, such as zinc depletion described in patients treated with DTPA [44].

EDTA activity has been evaluated most notably against *P. aeruginosa* and has been shown to decrease the formation of biofilms by this microorganism. In a study by Banin et al. [45], a concentration of 50 mM was able to reduce the number of cells associated with biofilms by >99%. Furthermore, this activity is inversely proportional to the concentration of iron or calcium in the medium, demonstrating that metal chelation is involved in its mechanism of action. In a separate study, higher concentrations of EDTA (30 mg/mL; approximately 100 mM) were required to reduce the cell count associated with biofilms by approximately 2 log_10_ [35]. Different laboratory conditions, as well as the *P. aeruginosa* strain used, may contribute to the difference observed. EDTA can significantly inhibit the growth of *P. aeruginosa* in both aerobic and anaerobic conditions at concentrations of 1250 µM and 650 µM, respectively, and impair the biofilm formation to 312 µM [36]. EDTA has also been combined with phenyl-arginine-β-naphthylamide (PAβN), an inhibitor of the MexAB-OprM efflux system in *P. aeruginosa* which is overexpressed when the bacterium grows under severe iron limiting conditions [46]. The combination of EDTA with 50 µg/mL PaβN is able to significantly inhibit planktonic growth at low concentrations (2.5 µg/mL), and even block it completely at 160 µg/mL [47]. The combination of 5 µg/mL EDTA with PaβN also inhibits the formation of biofilm in *P. aeruginosa* [47].

DTPA has also shown antimicrobial activity against different species in vitro. O’May et al. describe activity similar to EDTA in inhibiting the growth and biofilm formation of *P. aeruginosa* in both aerobic and anaerobic conditions at concentrations of 1250 µM and 650 µM, respectively [36]. DTPA can also inhibit the growth of *S. aureus* (bacterial reduction of 97.3% at 500 µg/mL), *P. aeruginosa* (bacterial reduction of 86.2% at 100 µg/mL) and *E. coli* (bacterial reduction of 98.5% at 250 µg/mL) [30].

### 3.3. Hydroxypyridinones

Hydroxypyridinones (HPO) (Figure 2C) are a group of synthetic compounds that have affinity for Fe(III) similar to that of cathecols, and the stability of the iron-chelator complexes are similar to that described for hydroxamates. However, unlike these two types of ligands, the chemical structure of HPOs does not resemble ligands that are found in the bacterial siderophores. This difference in structure makes Fe(III)-HPO complexes difficult to capture and exploit by pathogenic microorganisms, in contrast to, for example, DFO. Within the different types of HPO, the 3-hydroxypyridin-4-ones are the ligands of this group that have the highest affinity for Fe(III) [19].

Deferiprone (1,2-dimethyl-3-hydroxypyridin-4-one; DFP) is a bidentate ligand with high affinity for Fe(III) and has been approved for the treatment of iron overload in patients with thalassemia [48]. This compound has moderate activity in vitro against Gram positive and Gram negative species. The growth of *S. aureus* and other coagulase-negative staphylococci is inhibited in the presence of 1.5 mM DFP (approximately 200 µg/mL) after 6 h of incubation, although *S. aureus* regrowth at the same level as the control culture occurred after 24 h [49]. In a separate study by Thompson et al. the in vitro activity of DFP is evaluated against important nosocomial pathogens, demonstrating minimal inhibitory concentrations (MICs) of 128–512 µg/mL for *A. baumannii*, *E. coli*, *Klebsiella pneumoniae* and *P. aeruginosa* in cation-adjusted Mueller-Hinton broth. When the same assay is performed in the iron-poor culture medium RPMI 1640, the MICs are 2–4 fold lower in some cases. DFP is not able to inhibit the growth of *S. aureus* [24]. Consistent with these results, de Léséleuc et al. obtain similar results with *A. baumannii* in the presence of DFP with a MIC of 128 µM in media with high and low iron content [42]. Although the antimicrobial activity of DFP seems to be mainly related to its iron chelating capability, Visca et al. suggest that there must be a chelation-independent toxicity since the addition of iron reverse but does not prevent the antimicrobial activity at high concentrations of DFP (≥3.67 mM) [41].

As previously described for deferoxamine, DFP can act as a growth promoter for some bacterial species, like *P. aeruginosa* and *A. baumannii*, at sub-inhibitory concentrations in iron-limiting conditions [41,42]. Moreover, DFP can capture Fe(III) bound to proteins such as transferrin [50], and is able to mobilize it from these iron sources thus facilitating its uptake by *A. baumannii* [42].

Another HPO with antimicrobial activity is cyclopirox (6-cyclohexyl-1-hydroxy-4-methylpyridin-2-one), a topical antifungal used for decades in the treatment of superficial mycoses. Although its mechanism of action is not fully characterized, studies carried out with *Candida albicans* have revealed changes in the expression of genes similar to those that occur in conditions of iron limitation [51,52]. Furthermore, its activity decreases with the addition of iron to the culture [51,52]. In a study by Carlson-Banning et al., cyclopirox is able to inhibit the growth of *E. coli* (MICs 5–15µg/mL), *K. pneumoniae* (MICs 5–15µg/mL), *A. baumannii* (MICs 5–7 µg/mL), and *P. aeruginosa* (MICs 10 - >30 µg /mL) with differing antibiotic resistance profiles [27].

As mentioned above, hexadentate ligands have higher affinity for Fe(III) than bidentate ligands with similar chemical structures. For this reason, hexadentate compounds derived from HPO have been designed with the aim of achieving Fe(III) binding constants similar to that of bacterial siderophores in order to compete with them for iron sequestration. The hexadentate compound CP251 is able to completely inhibit the in vitro growth of *S. aureus* and *P. aeruginosa* at 500 µg/mL and 100 µg/mL, respectively [30]. It also decreases the growth of *E. coli* 2log_10_ at 250 µg/mL (bactericidal rate of 99.6%) [30]. Several studies have analyzed various hexadentate HPOs with results similar to Qiu et al. [53,54].

### 3.4. DIBI

The development of hexadentate HPOs has also focused on the design of larger compounds with less systemic absorption than smaller molecules in order to avoid some of the adverse effects derived from iron depletion in certain biological compartments. Polymers or dendrimers (branched polymers) derived from HPO may have significant antimicrobial activity and be effective in treating localized infections, such as wound infections or infections of the respiratory mucosa. One example of this type of compound is DIBI [55], a 9 kDa copolymer that has nine broadly spaced hydroxypyridinone metal-binding groups that can coordinate with three iron molecules with full hexadentate binding (Figure 3). DIBI is highly water soluble, shows selective binding to Fe(III) ions and has not demonstrated toxicity in preclinical animal models [55]. In recent years, multiple studies have demonstrated the antimicrobial activity of DIBI against multiple antibiotic resistant bacterial species including both Gram positive and Gram negative pathogens. Initial studies with DIBI demonstrated strong antimicrobial activity against antibiotic susceptible reference strains for *S. aureus* (MIC; 4 µg/mL), *A. baumannii* (MIC; 2 µg/mL) and *C. albicans* (MIC; 2 µg/mL). These MIC values were one to two log_10_ lower than MIC values observed for previously the characterized iron chelators DFP and DFO, respectively. Importantly, given the large size of DIBI compared to these chelators, the molecular weight of DIBI is 9 kDa vs. 139 Da for DFP, these MICs translate into very low concentrations of the polymer.

In a separate study, DIBI also demonstrated activity against methicillin resistant *S. aureus* isolates, with MICs between 1 and 4 µg/mL [32]. DIBI also showed activity in vivo against *S. aureus* in a wound infection model, resulting in reduced bacterial loads and tissue inflammation compared to control mice, and intranasal administration of DIBI was also able to reduce nasal *S. aureus* carriage by approximately two log_10_ compared to control mice [32]. In a separate study, DIBI was able to reduce nasal *S. aureus* carriage of both mupirocin susceptible and resistance isolates in a mouse model [33]. Importantly, no post-treatment resistance to DIBI was observed. DIBI demonstrates strong antimicrobial activity in vitro against clinical isolates of *A. baumannii*, with MIC values of 4 µg/mL [28]. Intranasal administration of DIBI was also able to reduce tissue bacterial loads, reduce dissemination to the spleen and improve survival in a mouse pulmonary infection model [28].

In addition to its antimicrobial activity, data from preclinical experimental models have indicated that DIBI has anti-inflammatory activity during sepsis [56,57]. Treatment with DIBI was able to reduce inflammation-associated markers (leukocyte activation, loss of capillary perfusion and tissue damage) after administration of lipotechoic acid from *S. aureus* and lipopolysaccharide from *E. coli* and *K. pneumoniae* [56]. Although the molecular mechanisms underlying the anti-inflammatory activity observed with DIBI are not fully understood, it has been hypothesized that is may be due to reduced iron-catalyzed reactive oxygen species production by leukocytes during sepsis [56,57]. Taken together, these studies indicate that DIBI may be a promising antimicrobial compound.

### 3.5. Other Iron Chelators

There are additional chelating compounds that can bind iron with high affinity. Some of the first molecules tested against microorganisms have a natural origin, as is the case for the human proteins lactoferrin and transferrin. Both compounds can inhibit the formation of biofilms in *P. aeruginosa* [58,59,60]. Their in vitro activity against *S. aureus*, *A. baumannii*, *K. pneumoniae* and *C. albicans* has also been described [61,62]. However, some pathogens, such as *P. aeruginosa*, are able to use these proteins as an iron source [63], which limits their efficacy as antimicrobials against these microorganisms. Compounds derived from cathecol have high affinity for Fe (III) due to the high electron density generated by the two oxygen atoms (Figure 2D) [20]. Cathecol moieties are a fairly common motif in bacterial siderophores. However, this electron charge density also gives them high affinity for protons, making these compounds less stable within the physiological pH range [20]. Compounds derived from 2,3-dihydroxybenzoic acid have been show to inhibit the formation of *P. aeruginosa* biofilms [64,65].

Deferasirox (DFX) is a triazole-derived trivalent chelator that was approved by the FDA in 2005 for the treatment of iron overload. DFX can inhibit antibiotic resistant strains of *S. aureus* with a MIC and a minimum bactericidal concentration of 50 mg/L [34]. It is also able to reduce in vitro biofilm formation of *P. aeruginosa* by 99% at a concentration of 1 µM after 4 h of treatment [25]. However, the use of DFX is limited by its toxicity [66].

The chelator 2,2′-bypiridyl (BIP), which can cross biological membranes and exerts its action in the cytoplasm of cells, has been widely studied in vitro. It should be noted that it has greater affinity for Fe(II) than for Fe(III). BIP inhibits the growth of pathogens such as *K. pneumoniae*, *E. coli*, *A. baumannii*, *S. aureus* and *P. aeruginosa* with MICs between 64 and 512 µg/mL [24,29], and can inhibit the formation of *P. aeruginosa* biofilm [36]. The therapeutic use of BIP is limited by its neurotoxicity [67]. However, this chelator is used in the laboratory to simulate iron-limiting conditions experimentally [41,68,69].

Finally, derivatives of 8-hydroxyquinoline have also been explored as antimicrobial iron chelators. An example of this type of compound is nitroxoline (5-nitro-8-hydroxyquinoline), a well-known antimicrobial used for the treatment of urinary tract infections. It can chelate both divalent and trivalent metals and this chelation is pH dependent [70]. Nitroxoline inhibits the planktonic growth of *P. aeruginosa*, *K. pneumoniae* and *E. coli* [31]. It can also reduce biofilm formation by up to 80% and decrease 4log_10_ the number of viable cells in preformed *P. aeruginosa* biofilms at concentrations of 8 and 200 µg/mL, respectively [31]. Another 8-hydroxyquinoline derivative is VK28, which demonstrates in-vitro activity against important nosocomial pathogens when the growth medium is poor in iron [24].

## 4. Gallium

The therapeutic properties of gallium, specifically Ga(III), are due to its physical-chemical similarity to Fe(III) (reviewed in [71,72]). Unlike Fe(III), Ga(III) cannot be reduced under physiological conditions. This inability to participate in redox reactions likely inhibits the catalytic activity of numerous enzymes and blocks essential functions of the bacterium. However, the specific molecular mechanisms of action responsible for the antimicrobial activity of Ga(III) have not been fully characterized. Recently, Wang et al. demonstrated Ga(III) binding to two subunits of RNA polymerase in *P. aeruginosa*, potentially explaining RNA synthesis inhibiting activities [73]. Due to its ability to alter iron metabolism, the antimicrobial potential of various compounds derived from gallium has been studied in multiple studies in recent years.

One of the first compounds tested was gallium nitrate (Ga(NO_3_)_3_), which was approved by the FDA for treatment of cancer-associated hypercalcemia. The activity of Ga(NO_3_)_3_ is moderate against *A. baumannii* regardless of the iron concentration of the medium [42,68]. However, the activity improves when serum is added to the medium or if it is cultivated directly on serum-containing solid media [42,68,74]. This effect is dose and strain dependent, and 90% growth inhibition is achieved at concentrations between 3 and 64 µM (Table 2). Ga(NO_3_)_3_ also inhibits the formation of *A. baumannii* biofilm in human serum at 16 µM, and can disrupt preformed biofilms at 64µM [74]. This increased activity could be due to a synergistic effect with transferrin, which may chelate Ga(III) and act as a platform for gaining access to the interior of bacterial cells. Furthermore, there is evidence that the presence of this protein can induce expression the iron uptake system in *A. baumannii* [75]. An increase in production of siderophores may facilitate the uptake of Ga(III), as seen in other species [76,77]. The use of Ga(NO_3_)_3_ has also been evaluated against *S aureus*, and is able to significantly inhibit the growth of planktonic cells and preformed biofilms at ≥16 µM and ≥128 µM, respectively [78].

Gallium-protoporphyrin IX (GaPPIX) is another compound that has been evaluated as an antimicrobial. This non-iron heme derivative uses heme-uptake pathways to reach the interior of the cell and inhibit hemoproteins such as cytochromes, catalases or peroxidases [85]. Hizaji et al. have described that GaPPIX inhibits the growth of *P. aeruginosa* (IC_50_ = 12 µM) by targeting heme-dependent *b*-type cytochromes [84]. GaPPIX also inhibits different strains of *A. baumannii* at concentrations ≤20 µM [81]. Based on the idea that the access route that GaPPIX exploits is different from that of Ga(NO_3_)_3_, Choi et al. evaluated the combined use of these compounds against different nosocomial pathogens [79]. The combination of GaPPIX and Ga(NO_3_)_3_ demonstrated a synergistic effect against *P. aeruginosa*, *A. baumannii*, *K. pneumoniae* and *S. aureus* with fractional inhibitory concentrations index (FICI) of 0.5, 0.5, 0.13 and 0.37, respectively; with an FICI of ≤0.5 considered as a synergistic effect between two compounds against a specific microorganism. Furthermore, this synergy was also reflected in the disruption of *K. pneumoniae* and *P. aeruginosa* biofilms (>90% and >95%, respectively, compared to the positive control) at concentrations in which they individually have no activity [79].

Other compounds derived from gallium such as gallium maltolate (GaM) and gallium citrate (GaCi) have been tested against bacteria such as *S. aureus* or *K. pneumoniae* [82,83]. In a study by Hijazi et al. the in-vitro activity of GaM, GaPPXI and Ga(NO_3_)_3_ against different antibiotic resistant nosocomial pathogens in different Fe(III) concentration and in a medium supplemented with human serum (RPMI-HS) were determined [80]. The susceptibility of these microorganisms to different gallium compounds varied between species and strains. Susceptibility was greater in iron-poor conditions (like RPMI medium supplemented with serum) with GaM and Ga(NO_3_)_3_, and GaPPIX was the only compound with activity in serum-free media, regardless of Fe(III) concentration. This may be explained by the different systems employed for heme-uptake in these species, and by the presence of albumin in serum which may block the action of GaPPIX [80].

Finally, gallium-derived compounds have also shown efficacy in vivo. Both Ga(NO_3_)_3_ and GaPPIX increased the survival of *Galleria mellonella* in models of *A. baumannii* infection [68,81]. Ga(NO_3_)_3_ is also active in models of *P. aeruginosa* and *A. baumannii* lung infection in mice [42,86]. Moreover, there is a phase 1b human trial whose results have been published recently [87]. In this study, the antibiotic activity of gallium in patients with cystic fibrosis and chronic *P. aeruginosa* airway infections was evaluated. Ga(NO_3_)_3_ was administered by slow intravenous infusion over 5 days in two different dose regimens (100 and 200 mg/m^2^/day). No serious adverse effects were observed, and plasma and sputum gallium levels remained detectable for prolonged periods. Remarkably, these patients presented a statistically significant increase in lung function in a magnitude similar to that produced by approved antimicrobials in cystic fibrosis. In addition, Ga(NO_3_)_3_ treatment decreased sputum *P. aeruginosa* concentrations, but they were not statistically significant. There are two other human trials in which gallium antimicrobial activity has been evaluated in cystic fibrosis patients: a phase 2 study for patients with cystic fibrosis infected with *P. aeruginosa* (IGNITE study) and a phase 1 study for patients with cystic fibrosis who are colonized with nontuberculous mycobacterias (ABATE study) (www.clinicaltrials.gov accessed on 9 February 2021).

## 5. Iron Chelators and Gallium Combinations

As previously seen, some iron chelators might act as carriers for other antimicrobial compounds within bacteria. In this way, Ga(III) has also been tested in combination with deferoxamine. DFO is captured by bacterial species such as *S. aureus* or *P. aeruginosa*, potentially acting as a Ga(III) carrier. DFO-Ga(III) can inhibit the planktonic growth of *P. aeruginosa* [26] and *E. coli* [43] better than the Ga(III) alone; while it does not appear to have activity against *S. aureus* [43]. Moreover, DFO-Ga(III) blocks *P. aeruginosa* biofilm formation at planktonic subinhibitory concentrations (0.001 mM) and cause a 3–4 log_10_ decrease in cell count of *P. aeruginosa* established biofilms [26].

Another combined strategy has been the use of DFP together with GaPPIX. These two compounds applied sequentially (first DFP and then GaPPIX) significantly reduced preformed *S. aureus* biofilms compared to the compounds used separately (94% vs 77% of de most active single compound) [88]. However, in another study by Richter et al., the formulation of DFP and GaPPXI in gel form failed to have a greater effect against *S. aureus* biofilm compared to GaPPXI alone [89]. In this case, there was a significant overall effect against a *P. aeruginosa* biofilm (2 log_10_ reduction compared to GaPPXI alone) [89].

## 6. Iron Chelators and Gallium in Combination with Antimicrobials

As described previously, alteration of iron homeostasis has a deleterious effect on bacterial growth. In addition to this inhibition per se, a decrease in intracellular iron concentrations can enhance the activity of other antimicrobials. Thus, combined treatment with currently-used antibiotics and iron chelators or Ga(III) compounds could have synergistic effects.

Some iron chelators such as DFO or DFX have been evaluated together with other antimicrobials [25,34]. DFX can increase vancomycin binding to the cell surface and together they have greater activity in vitro and in vivo against *S. aureus* than each compound used separately [34]. In a study by Moreau-Marquis et al. the combined use of DFO or DFX with tobramycin decreased the biomass of preformed *P. aeruginosa* biofilms by 90% [25]. DFX and tobramycin used together also significantly decrease the formation of *P. aeruginosa* biofilms [25].

Compounds derived from HPO also increase the activity of other antimicrobials. One example is DIBI, the copolymer derived from 3-hydroxypyridin-4-one [55]. This compound has shown in-vitro synergy with multiple antimicrobials such as gentamicin and ciprofloxacin against *A. baumannii* and *S. aureus*, including isolates resistant to the co-administered antibiotic [28,32,33].

There are also several studies testing Ga(III) derived compounds together with other antimicrobials. Antunes et al. demonstrated that Ga(NO_3_)_3_ acts synergistically with colistin against *A. baumannii* (FICI values ranging from 0.13 to 0.5) [68]. The alteration of the outer membrane produced by colistin probably favors the diffusion of Ga(III) into the cell [68]. Another study also describes synergy between Ga(NO_3_)_3_ and GaPPIX together with colistin, rifampin or ciprofloxacin against *K. pneumoniae*, *P. aeruginosa* and *S. aureus* [79]. Finally, the gel formulation of DFP and GaPPXI effective against *P. aeruginosa* biofilms when used alone demonstrated improved activity when used together with ciprofloxacin [89].

## 7. Challenges to Developing Iron Chelation/Gallium-Based Therapies

Although iron chelation and gallium-based therapies have demonstrated antimicrobial activity in-vitro and in a handful of in-vivo studies, multiple challenges are associated with the development of these therapies for clinical use. First, during preclinical development of these compounds, the antimicrobial activity observed with iron chelators and gallium compounds is highly dependent upon the conditions used for their characterization. Given that these compounds target bacterial iron metabolism, the concentration of free iron in the assays used for quantifying activity can significantly affect results. As can be appreciated in Table 1, multiple assay conditions have been employed, most typically bacterial growth medium (such as Mueller-Hinton broth) and cell culture media (RPMI). Some of these assay conditions may not reflect physiologically relevant iron concentrations, thus altering the observed antimicrobial activities. This may be especially true for iron rich laboratory growth media. As described in the preceding sections, growth media can significantly influence the observed antimicrobial activity of iron chelating and gallium compounds. In addition to iron, concentrations of other polyvalent cations such as zinc and manganese, can potentially affect the activity since it is known that some chelators have high affinity for these metals. For these reasons, the use of media that accurately reflect physiological ion concentrations may be desirable in initial in-vitro assays assessing chelator or gallium activity. An additional important consideration is the strain-dependent variation of the activity of iron chelators and gallium-based compounds. Initial in-vitro studies characterizing the activity these compounds often employ antibiotic susceptible reference strains, which may not be representative of antibiotic resistant clinical isolates that produce disease. Many of the studies included in this review have characterized activity against antibiotic resistant isolates [23,24,28,32,33,68,80], a critical aspect required for appropriately evaluating the therapeutic potential of these compounds. In-vivo studies can provide highly relevant information on chelator/gallium activity as these models are physiological with respect metal ion concentrations, and also account for aspects such as compound biodistribution, pH and the presence of host factors.

A second aspect that must be addressed for iron chelators and gallium-based compounds is elucidating the mechanisms by which they affect existing antimicrobials. One of the most likely uses of these compounds is in combination therapy together with existing antibiotics; however, very little is known regarding how chelators and gallium interact with currently approved antimicrobials. The activity seen with some combinations of chelators/gallium and antibiotics (described above) may reflect the summative effects of the antibiotic with reduced bacterial growth/survival in the absence of available iron. However, there is evidence indicating that the additive effects of these chelator/antibiotic combinations may be more complex. Free iron concentrations can alter bacterial transcriptional programs, most notably through the Fur protein [90]. Under low iron concentrations, Fur-mediated transcriptional repression is decreased resulting in multiple physiologic changes in the bacteria, including a switch to planktonic growth [45,91]. Planktonic bacteria are known to be more susceptible to certain antibiotics, indicating a possible mechanism of iron chelation-induced sensitization to antibiotics. Additional study is needed in order to fully elucidate how altering bacterial iron metabolism can affect antibiotic susceptibility.

A final aspect that must be fully addressed is the potential for toxicity with the use of iron chelating and gallium-based compounds. The polyvalent cations that bind with high affinity to these compounds are essential for multiple host physiological processes, raising the possibility that altering their concentration/metabolism could have adverse effects on host cells and tissues. Although some iron chelators have been approved for clinical use in the treatment of non-infectious indications, toxicity associated with iron chelation therapy has been described [92,93]. Future study is needed in order to characterize the potential toxicity of different iron chelating and gallium compounds at concentrations used for the treatment of bacterial infections.

## 8. Conclusions and Future Directions

The absolute requirement for iron in multiple essential bacterial processes of many pathogenic bacteria has raised interest in the use of iron chelators and gallium compounds as antimicrobial therapies over the previous decade. The majority of studies assessing these compounds have characterized their antimicrobial activity in vitro. Although multiple compounds have demonstrated promising results against some of the most important multidrug resistant strains, studies characterizing the antimicrobial activity of these compounds under physiological conditions are needed in order to fully evaluate their potential use as therapeutics. In-vivo studies in animal models are of special interest as they account for factors such as biodistribution and the presence of host factors, aspects which cannot be adequately addressed in most in vitro assays. Studies describing potential synergistic effects of iron chelators and gallium compounds when used with existing antibiotics indicate that combination therapy may be an effective therapeutic approach. Additional work is needed to understand the mechanisms by which iron chelators and gallium potentiate the activity of antibiotics.

## Figures and Tables

**Figure 1 ijms-22-02876-f001:**
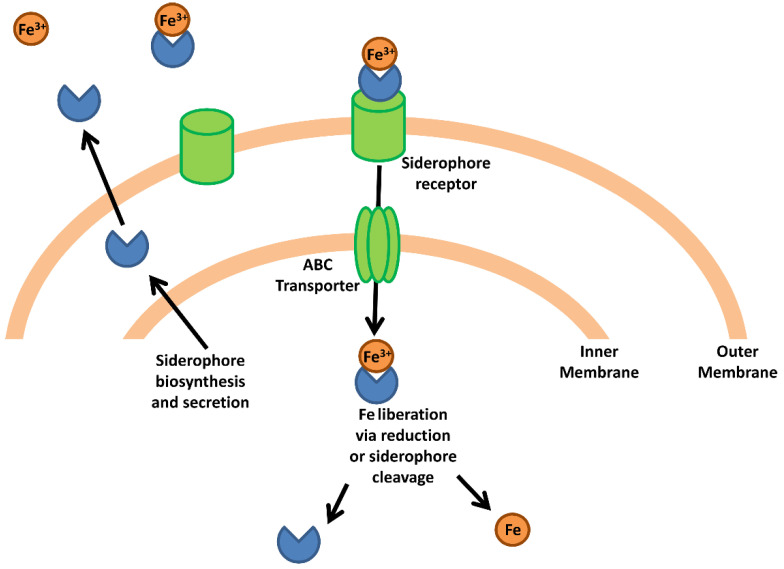
Bacterial iron acquisition via siderophores. The schematic represents a generic siderophore-mediated iron acquisition system based on common features in Gram-negative species. Siderophores are synthesized and secreted from the bacterial cell where they capture free iron or strip iron complexed to host proteins. The siderophore-iron complex binds a siderophore-specific receptor on the bacterial cell surface and is transported through the extracellular membrane. An ABC transporter transports the siderophore-iron complex into the cytoplasm where the complex is dissociated via reduction or enzymatic cleavage of the siderophore, releasing the iron atom into the intracellular iron pool.

**Figure 2 ijms-22-02876-f002:**
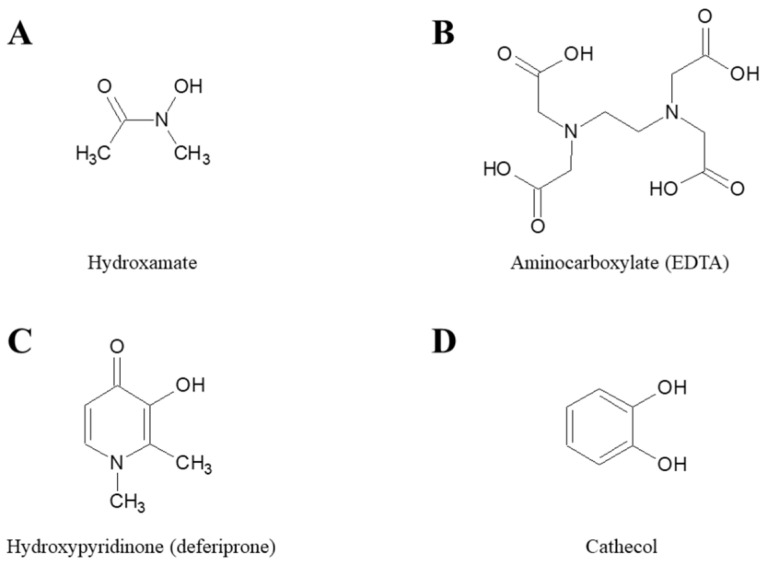
Chemical structures of the parent molecules of different classes of iron chelators. (**A**) Hydroxamate, (**B**) Aminocarboxylate, (**C**) Hydroxpyridinone, and (**D**) Cathecol.

**Figure 3 ijms-22-02876-f003:**
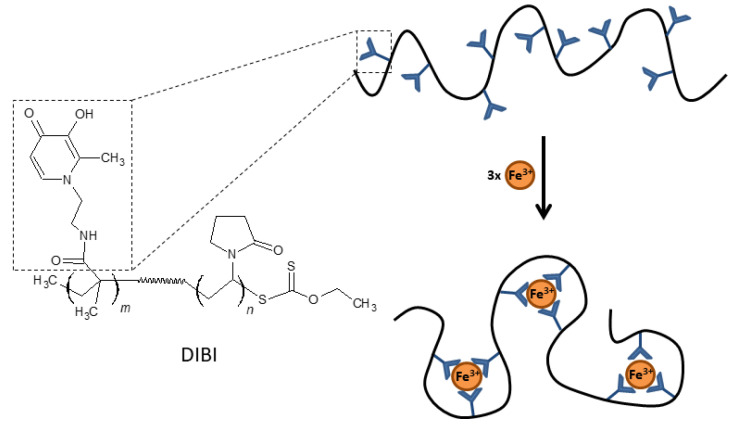
Chemical structure and schematic representation of Fe(III) binding by DIBI.

**Table 1 ijms-22-02876-t001:** In vitro antimicrobial activity of iron chelators.

Microorganism	Effect on	Compound	Growth Condition	Compound Inhibitory Concentration	Ref.
*Acinetobacter baumannii*	Planktonic growth	DFO	MH broth	MIC = 2.5–10 mg/mL	[23]
			CAMH broth	MIC > 512 µg/mL	[24]
			RPMI 1640	MIC > 512 µg/mL	[24]
		Exo-MS	MH broth	MIC = 0.05–0.25 mg/mL	[23]
		DFP	CAMH broth	MIC = 128 µg/mL	[24]
			RPMI 1640	MIC = 64–128 µg/mL	[24]
		Ciclopirox	MH broth	MIC = 5–7 µg/mL	[27]
		DIBI	RPMI 1640	MIC = 2 µg/mL (0.2µM)	[28]
		BIP	CAMH broth	MIC = 32–4 µg/mL	[24,29]
			RPMI 1640	MIC = 32 µg/mL	[24]
		VK28	CAMH broth	MIC = 128 µg/mL	[24]
			RPMI 1640	MIC = 8–32 µg/mL	[24]
*Escherichia coli*	Planktonic growth	DFO	CAMH broth	MIC > 512 µg/mL	[24]
			RPMI 1640	MIC > 512 µg/mL	[24]
		DTPA	BHI broth	IC_90_ = 250 µg/mL	[30]
		DFP	CAMH broth	MIC = 512 µg/mL	[24]
			RPMI 1640	MIC = 256–512 µg/mL	[24]
		Ciclopirox	MH broth	MIC = 5–15 µg/mL	[27]
		CP251	BHI broth	MBC = 250 µg/mL	[30]
		BIP	CAMH broth	MIC = 64 µg/mL	[24]
			RPMI 1640	MIC = 64 µg/mL	[24]
		Nitroxoline	MH broth	MIC = 2 µg/mL	[31]
		VK28	CAMH broth	MIC > 512 µg/mL	[24]
			RPMI 1640	MIC = 8–32 µg/mL	[24]
*Klebsiella pneumoniae*	Planktonic growth	DFO	CAMH broth	MIC > 512 µg/mL	[24]
			RPMI 1640	MIC > 512 µg/mL	[24]
		DFP	CAMH broth	MIC = 256–512 µg/mL	[24]
			RPMI 1640	MIC = 256 µg/mL	[24]
		Ciclopirox	MH broth	MIC = 5–15 µg/mL	[27]
		BIP	CAMH broth	MIC = 256–512 µg/mL	[24]
			RPMI 1640	MIC = 128–256 µg/mL	[24]
		Nitroxoline	MH broth	MIC = 4 µg/mL	[31]
		VK28	CAMH broth	MIC > 512 µg/mL	[24]
			RPMI 1640	MIC = 16 µg/mL	[24]
*Staphylococcus aureus*	Planktonic growth	DFO	MH broth	MIC = 7.5–10 mg/mL	[23]
			CAMH broth	MIC > 512 µg/mL	[24]
		DTPA	BHI broth	IC_90_ = 500 µg/mL	[30]
		Exo-MS	MH broth	MIC = 0.05–0.5 mg/mL	[23]
		DFP	CAMH broth	MIC > 512 µg/mL	[24]
		CP251	BHI broth	MBC = 500 µg/mL	[30]
		DIBI	RPMI 1640	MIC = 1–4 µg/mL (0.1–0.4µM)	[32]
			RPMI 1640	MIC = 2–8 µg/mL (0.22–0.88µM)	[33]
		DFX	MH broth	MIC y MBC = 50 mg/L	[34]
		BIP	CAMH broth	MIC = 256–512 µg/mL	[24]
		VK28	CAMH broth	MIC = 256 µg/mL	[24]
			RPMI 1640	MIC = 16–32 µg/mL	[24]
*Pseudomonas aeruginosa*	Planktonic growth	DFO	MH broth	MIC = 5–10 mg/mL	[23]
			CAMH broth	MIC > 512 µg/mL	[24]
			RPMI 1640	MIC > 512 µg/mL	[24]
		Exo-MS	MH broth	MIC = 0.125 mg/mL	[23]
		EDTA	LB	MIC = 6250 µg/mL	[35]
		DFP	CAMH broth	MIC = 256–>512 µg/mL	[24]
			RPMI 1640	MIC = 128–512 µg/mL	[24]
		Ciclopirox	MH broth	MIC = 10–>30 µg/mL	[27]
		CP251	BHI broth	IC_90_ = 100 µg/mL	[30]
		BIP	CAMH broth	MIC = 256 µg/mL	[24]
			RPMI 1640	MIC = 256 µg/mL	[24]
		Nitroxoline	MH broth	MIC = 16–64 µg/mL	[31]
		VK28	CAMH broth	MIC > 512 µg/mL	[24]
			RPMI 1640	MIC = 16 µg/mL	[24]
	Biofilm growth	DFO	Flow chamber (CFBE cells with MEM)	BIC = 400 µg/mL (0.71 mM)	[25]
		EDTA	Borosilicate glass tube (LBN)	BIC = 312 µM	[36]
		DTPA	Borosilicate glass tube (LBN)	BIC = 625–1250 µM	[36]

Abbreviations: BHI, brain-heart infusion; BIC, biofilm inhibitory concentration; BIP, 2,2′-bypiridyl; CAMH, cation-adjusted mueller-hinton; CFBE, modiefied human bronchial ephitelial cells [25]; DFO, deferoxamine; DFP, deferiprone; DFX, deferasirox; DTPA, diethylenetriaminepentaacetic acid; Exo-Ms, exochelin-MS; IC_90_, compound concentration that inhibit grown by 90% LB, Luria-Bertani broth; LBN, Luria-Bertani broth plus nitrate; MBC, minimum bactericidal concentration; MEM, supplemented minimum essential medium [25]; MHB, Mueller-Hinton; MIC, minimum inhibitory concentration; RPMI, Roswell Park Memorial Institute.

**Table 2 ijms-22-02876-t002:** In-vitro antimicrobial activity of Gallium compounds.

Microorganism	Effect on	Compound	Growth Condition	Compound Inhibitory Concentration	Ref.
*Acinetobacter baumannii*	Planktonic growth	Ga(NO_3_)_3_	M9-DIP	IC_90_ = 2–80 µM	[68]
			HS	IC_90_ = 4–64 µM	[68]
			HS	IC_90_ = 3.1 µM	[42]
			HS	IC_90_ = 3.8–31 µM	[74]
			BM2	MIC = 4 µg/mL	[79]
			DMHB	MIC > 128 µM	[80]
			RPMI-HS	MIC = 1–2 µM	[80]
		GaM	DMHB	MIC > 128 µM	[80]
			RPMI-HS	MIC = 1 µM	[80]
		GaPPIX	CAMH broth	MIC = 20 µM	[81]
			BM2	MIC = 4 µg/mL	[79]
			DMHB	MIC = 16–32 µM	[80]
			RPMI-HS	MIC = 0.25–128 µM	[80]
	Biofilm growth	Ga(NO_3_)_3_	Microtitre plates (HS)	BIC = 16 µM	[74]
*Escherichia coli*	Planktonic growth	DFO-Ga	TSB 1%	IC_50_ = 42 µM	[43]
*Klebsiella pneumoniae*	Planktonic growth	Ga(NO_3_)_3_	BM2	MIC = 4 µg/mL	[79]
			DMHB	MIC > 128 µM	[80]
			RPMI-HS	MIC = 4–>128 µM	[80]
		GaM	DMHB	MIC > 128 µM	[80]
			RPMI-HS	MIC = 2–>128 µM	[80]
		GaCi	BM2	MIC = 0.125–2 µg/mL	[82]
		GaPPIX	BM2	MIC = 16 µg/mL	[79]
			DMHB	MIC > 128 µM	[80]
			RPMI-HS	MIC > 128 µM	[80]
*Staphylococcus aureus*	Planktonic growth	Ga(NO_3_)_3_	BM2	MIC = 512 µg/mL	[79]
			DMHB	MIC > 128 µM	[80]
			RPMI-HS	MIC > 128 µM	[80]
		GaM	RPMI	MIC = 375–2000 µg/mL	[83]
			DMHB	MIC > 128 µM	[80]
			RPMI-HS	MIC = 128–>128 µM	[80]
		GaPPIX	CAMH broth	MIC = 0.031–0.062 µg/mL	[79]
			DMHB	MIC = 0.06–0.12 µM	[80]
			RPMI-HS	MIC > 128 µM	[80]
		DFO-Ga	TSB 1%	IC_50_ = 565 µM	[43]
	Biofilm growth	GaM	Microtitre plates (RPMI)	MBIC = 3000–>6000 µg/mL	[83]
*Pseudomonas aeruginosa*	Planktonic growth	Ga(NO_3_)_3_	DCAA	IC_90_ = 12.5 µM	[84]
			BM2	MIC = 1–2 µg/mL	[79]
			DMHB	MIC = 64–>128 µM	[80]
			RPMI-HS	MIC = 0.5–16 µM	[80]
		GaM	DMHB	MIC > 128 µM	[80]
			RPMI-HS	MIC = 0.5–8 µM	[80]
		GaPPIX	DCAA	IC_50_ = 12.5 µM	[84]
			BM2	MIC = 8 µg/mL	[79]
			DMHB	MIC > 128 µM	[80]
			RPMI-HS	MIC = 8–128 µM	[80]
		DFO-Ga	TSB 1%	MIC = 0.032 mM	[26]
			TSB 1%	IC_50_ = 103 µM	[43]
	Biofilm growth	DFO-Ga	Flow cells (TSB 1%)	BIC = 0.001 mM	[26]
			Flow cells (TSB 1%)	BBC = 1 mM	[26]

Abbreviations: BBC, biofilm bactericidal concentration; BIC, biofilm inhibitory concentration; BM2, BM minimal medium with succinate; CAMH, cation-adjusted mueller-hinton; DCAA, iron-free Casamino Acids medium; DFO-Ga, deferoxamine-gallium; DMHB, iron-poor mueller-hinton broth; GaCi, gallium citrate; GaM, gallium maltolate; Ga(NO_3_)_3_, gallium nitrate; GaPPIX, gallium-protoporphyrin IX; HS, complement-free human serum; IC_50_, compound concentration that inhibit grown by 50%; IC_90_, compound concentration that inhibit grown by 90%; MBIC, minimum biofilm inhibitory concentration; MIC, minimum inhibitory concentration; M9-DIP, M9 minimal medium with 100 µM 2,2′-dipyridyl; RPMI, Roswell Park Memorial Institute; TSB, tryptic soy broth.

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
