# Peer review of "Recent Advances in Iron Chelation and Gallium-Based Therapies for Antibiotic Resistant Bacterial Infections"

_ijms, 2021, doi:10.3390/ijms22062876_

Round 1

Reviewer 1 Report

Review Comments for “Recent advances in iron chelation and gallium-based therapies for antibiotic resistant bacterial infections”

Overall Impression:

Paper does a good job presenting the literature, though some further support for background and introductory statements is needed. In addition, “antibiotic resistant” bacterial infections aren’t explicitly discussed in many cases, it would be appropriate for the authors to include literature or discussion of the resistance of the strains discussed as this is the focus of the title and introduction.

Minor Issues:

Typos:

Page 2 Line 74: Authors state concentration of iron in the body is 10-24 M, that seems impossibly high given iron isn’t even soluble in water above like 6 M.

Page 4 Linge 154: Extra space between the words “inhibit” and “the”.

Page 12 Table Legend: Many words are accented, they should not be.

Page 12 Line 367 and Page 13 Line 416: Abbreviation “FICI” is not defined, nor are the values reported with units for this measure, so it’s not at all meaningful.

Page 14 Line 481: Again, an extra space between the sentence ending “…assays.” and the word “Studies”.

Throughout: Mentions of papers in the text are given with the author name italicized ex.: “Moreau-Marquis et al.” I have to plead ignorance as to IJMS’s style guidelines, if that’s the recommendation then keep it, but I’ve always seen the author names not italicized so it struck me as odd.

Major Issues:

Antibiotic resistance:

The authors present in the title and their abstract this paper as a discussion of the utility of these classes as potential treatments for antibiotic resistant bacteria, but they don’t discuss explicitly whether a lot of the references presented are working on antibiotic resistant bacteria. Some discussion of this seems essential, especially given the discussion, in several places, of strain specific variation amongst the literature reports of the susceptibility of these bacteria.

Further referencing is needed of some background information:

The authors assume too much familiarity of the readership on this area some statements of fact are given without explicit referencing and that detracts from the utility of this report, especially for readers who are new to the area. As an example:

Page 9 Line 318: “However, this chelator is used in the laboratory to stimulate iron-limiting conditions experimentally.” Does not include a reference to reports using BIP this way, the authors should include some.

Other examples without references:

Page 6 Line 169: “However, the ability of DFO to be taken up by bacteria that do not produce it has been revealed as an effective strategy to transport different molecules with the antimicrobial activity into these pathogens.”

Page 3 and 4: The discussion of iron coordination chemistry.

Page 1: The discussion in the first few sentences of the Introduction could use some citations demonstrating the “significant burden on public health.” Or the affects mentioned on clinical management issues.

Reviewer 2 Report

In the Review entitled “Recent advances in iron chelation and gallium-based therapies for antibiotic resistant bacterial infections” the authors describe the limits and the potential of iron chelators and gallium as antimicrobials. The review is well written, but some essential articles have not been included in the review. I suggest the publication  after an extensive revision of the text.

Major revisions

Lines 86-90: Authors describe the different strategies employed by pathogenic bacteria to acquire ferric iron (Fe3+) from environment, but they do not mention that bacteria (almost all bacteria) are also able to acquire iron in the ferrous form (Fe2+, reviewed in Cheryl et al., 2015. Bacterial ferrous iron transport: the Feo system. FEMS Microbiology Reviews). Please add to the text the ferrous uptake system amongst the mechanisms used by bacteria to acquire iron.

Lines 96-97: Authors write “Upon reaching the bacterial cytosol, ferric iron is released to the intracellular iron pool either through cleavage of the siderophore or through the reduction of Fe(III) to Fe(II), which lowers binding affinity to the siderophore thus permitting its dissociation”. This sentence cannot be applied to all siderophores. Indeed, the human pathogen P. aeruginosa produces the siderophore pyoverdine which release iron in the periplasm and not in the cytoplasm (Imperi F, Tiburzi F, Visca P. Molecular basis of pyoverdine siderophore recycling in Pseudomonas aeruginosa. Proc Natl Acad Sci U S A. 2009 Dec 1;106(48):20440-5. doi: 10.1073/pnas.0908760106.). Please modify the text with an appropriate sentence.

Lines 116-118: Authors write “In the case of heme iron, multiple secreted bacterial effectors have been shown to lyse erythrocytes in order to liberate heme and its associated ferrous (Fe(II)) iron. Specific receptors on the bacterial cell surface bind these heme complexes which are then transported to the cytosol”. Beside the receptors on the bacterial cells capable to bind Heme, many bacterial pathogens secrete hemophores which take up heme from hemoglobin and shuttles it to the specific receptor on bacterial membrane. Please modify the text with an appropriate sentence.

In the paragraph 3.1, the effect of deferoxamine (DFO) against several pathogen was described. In particular, the authors write that the MIC of DFO against Staphylococcus aureus, Pseudomonas aeruginosa and Acinetobacter baumannii vary between 2.5 and 10 mg/ml in Mueller Hinton Broth (lines 154-157). The antibacterial activity of DFO was also investigated by Thompson and coworkers (Thompson MG, Corey BW, Si Y, Craft DW, Zurawski DV. Antibacterial activities of iron chelators against common nosocomial pathogens. Antimicrob Agents Chemother. 2012 Oct;56(10):5419-21. doi: 10.1128/AAC.01197-12.). They demonstrated that DFO up to 512 µg/ml did not inhibit P. aeruginosa, K. pneumoniae, A. baumannii, E. coli and S. aureus growth in Mueller Hinton Broth. The authors should add this work in the review.

In the second part of the same paragraph (lines: 164-167) the authors highlight that DFO may be used as xenosiderophore by P. aeruginosa, E. coli and S: aureus. However, they do not explain in which conditions and the concentration at which DFO may be exploit by the pathogens as iron carrier. The authors should mention the work “Visca P, Bonchi C, Minandri F, Frangipani E, Imperi F. The dual personality of iron chelators: growth inhibitors or promoters? Antimicrob Agents Chemother. 2013 May;57(5):2432-3. doi: 10.1128/AAC.02529-12.”. In this work, Visca and coworkers demonstrated that P. aeruginosa siderophore null mutant can exploit DFO (concentration tested 20µM) to acquire iron when cultivated in iron poor media. Authors should add this reference. Moreover, De Leseleuc et al., demonstrated that DFO in extreme iron limited condition such as those imposed by RPMI with human serum, promotes the growth of A. baumannii. This work is cited in the review (Lines223-225) but no data are reported concerning the effect of DFO on A. baumannii growth. The authors have to discuss in depth the effect of DFO on bacterial growth.

Lines 157-158: Authors state that DFO cause a reduction of 42% of biofilm formation in P. aeruginosa. However, they do not describe what is the concentration of DFO which inhibits P. aeruginosa biofilm. The efficacy of DFO against P. aeruginosa biofilm is controversial since Banin and collegues (Banin E, Lozinski A, Brady KM, Berenshtein E, Butterfield PW, Moshe M, Chevion M, Greenberg EP, Banin E. The potential of desferrioxamine-gallium as an anti-Pseudomonas therapeutic agent. Proc Natl Acad Sci U S A. 2008 Oct 28;105(43):16761-6. doi: 10.1073/pnas.0808608105.) clearly demonstrate (Figure 1D) that DFO alone is not able to inhibit P. aeruginosa biofilm. Please discuss in depth the effect of DFO on P. aeruginosa biofilm by taking into account also the article cited above. Regarding the effect of chelators on P. aeruginosa biofilm, in the table 1 the DFO is not cited. Please add this information in table 1.

In the paragraph 3.3 authors focus on the antibacterial activity of the chelator DFP, however they do not consider the work of Visca and coworker (Visca P, Bonchi C, Minandri F, Frangipani E, Imperi F. The dual personality of iron chelators: growth inhibitors or promoters? Antimicrob Agents Chemother. 2013 May;57(5):2432-3. doi: 10.1128/AAC.02529-12) which demonstrated that under iron limited condition DFP is able to promote P. aeruginosa growth (Figure 1A). The authors should discuss this date in the text.

In the paragraph 4, authors describe the antibacterial proprieties of gallium. In particular they highlighted that the anti-A. baumannii effect of the iron mimetic strongly depend on the iron availability in the assay medium. This is true not only for A. baumannii since gallium is an iron-mimetic and its antibacterial activity depends on the iron concentration in the test medium. Moreover Hijazi and coworker (Hijazi S, Visaggio D, Pirolo M, Frangipani E, Bernstein L, Visca P. Antimicrobial Activity of Gallium Compounds on ESKAPE Pathogens. Front Cell Infect Microbiol. 2018 Sep 10;8:316. doi: 10.3389/fcimb.2018.00316.) clearly demonstrate that Ga(NO3)3 and gallium maltolate is more effective against several bacterial species in condition of iron scarcity.

In the section 4 (Lines 382-392) the authors describe the antimicrobial activity of Ga(III) in combination with iron chelators. Since the review focus on Ga(III) and chelators, the description of the efficacy of the combination of both the antimicrobial, should be discussed in a separate section.

In the last part of paragraph 4, authors mention the efficacy of Ga(III) compounds in protecting Galleria mellonella and mice against P. aeruginosa and A. baumannii infections. However, the authors do not describe the result of a phase 1 human trial in which the antibiotic activity of gallium in people with cystic fibrosis (CF) chronic P. aeruginosa airway infections was investigated. Moreover, others clinical trials in which antibacterial activity of Ga(III) was evaluated are available on the web site www.clinicaltrails.gov (such as A Phase 2 IV Gallium Study for Patients With Cystic Fibrosis (IGNITE Study) and IV Gallium Study for Patients With Cystic Fibrosis Who Have NTM (ABATE Study)). Since the title of the review is “Recent advances in iron chelation and gallium-based therapies for antibiotic resistant bacterial infections” I think that the description of the published clinical trials (Goss CH, Kaneko Y, Khuu L, Anderson GD, Ravishankar S, Aitken ML, Lechtzin N, Zhou G, Czyz DM, McLean K, Olakanmi O, Shuman HA, Teresi M, Wilhelm E, Caldwell E, Salipante SJ, Hornick DB, Siehnel RJ, Becker L, Britigan BE, Singh PK. Gallium disrupts bacterial iron metabolism and has therapeutic effects in mice and humans with lung infections. Sci Transl Med. 2018 Sep 26;10(460):eaat7520. doi: 10.1126/scitranslmed.aat7520.) is mandatory.

Minor revisions

Line 74-75: Change “10-24 M” with “10-24 M”

Line 224: Change “De Leseluc” with “De Leseleuc”

Reviewer 3 Report

The paper titled "Recent advances in iron chelation and gallium-based therapies for antibiotic resistant bacterial infections" by Vinuesa and McConnell summarized the antimicrobial activity of iron chelators and gallium compounds. The authors focused on introducing studies related to novel iron chelator DIBI. Also the authors summarized the limitations especially in preclinical models. Overall, this review provides constructive information to this field however, there are several major issues:

Overall, the English needs to be improved significantly. Some sentences are quite confusing and misleading, e.g. "A report commissioned by the government of the United Kingdom in 2014 estimated that global deaths due to an- timicrobial resistance could increase to 10 million per year, compared to an estimated 700,000 deaths in 2014, if current trends continue". There are also bunch of grammar errors such as single vs plural.

The content is generally acceptable which also included recent researches. Only minor revision is needed:

Some minor issue:
1. Line 74, the authors mentioned free iron in human body is maintained at "a very low concentration". What's the standard here? It's lower than what?
2. Check Figure 1 the label "Fe" in a orange circle at the bottom after "Fe liberation via reduction or siderophore cleavage".
3. Line 185, check the format of number "2log10"

Round 2

Reviewer 2 Report

The revised version of the manuscript has been amended as suggested. All major raised issues have been taken into consideration and the novel information provided are fully satisfying. I now have no further comments, and I can recommend the publication.  

Reviewer 3 Report

The manuscript has been well improved and all my concerns are reasonably addressed. I believe this paper is acceptable for publication in IJMS.